# Dishes Adapted to Dysphagia: Sensory Characteristics and Their Relationship to Hedonic Acceptance

**DOI:** 10.3390/foods10020480

**Published:** 2021-02-23

**Authors:** Gorka Merino, Maria Remedios Marín-Arroyo, María José Beriain, Francisco C. Ibañez

**Affiliations:** Research Institute on Innovation & Sustainable Development in Food Chain (IS-FOOD), Universidad Pública de Navarra (UPNA), 31006 Pamplona, Spain; gorka.merino@unavarra.es (G.M.); remedios.marin@unavarra.es (M.R.M.-A.); mjberiain@unavarra.es (M.J.B.)

**Keywords:** adapted dish, texture-modified, check-all-that-apply, acceptability, sensory characteristics, dish identification

## Abstract

Dishes whose texture has been modified for dysphagia undergo changes in other sensory characteristics as well. Therefore, it is necessary to identify these characteristics in adapted dishes and their relationship to hedonic acceptance. In the present work, the sensory characteristics of five dishes adapted to dysphagia associated with cerebral palsy were investigated using the check-all-that-apply method. A hedonic evaluation with a panel of non-dysphagic judges was performed to relate the degree of acceptance with the sensory characteristics of the adapted dishes. The identification of the original non-adapted dish as well as the relationship between the hedonic evaluation by non-dysphagic judges and dysphagic judges were explored. The main attributes of the dishes adapted to dysphagia were “homogeneity” and “easy-to-swallow”. Attributes that increased the hedonic evaluation were “flavorful”, “flavor of the original dish”, “soft texture”, “easy-to-swallow”, and “odor of the original dish”. The attributes that decreased the hedonic evaluation were “thick mash” and “bland”. The fish dish was the only one correctly identified more than 62.5% of the time. The adapted dishes received scores above 4.7 out of 9.0 in the hedonic evaluation. The most accepted dishes were the chicken stew and the chickpea stew. Except for the pasta dish, the test yielded similar results to those obtained with dysphagic judges. The texture-modified dishes were correctly characterized and accepted. This study shows that all the sensory characteristics of the adapted dishes are crucial for acceptance and identification.

## 1. Introduction

Dysphagia is the disorder of normal liquid or solid food swallowing. It is associated with different causes, such as age or cerebrovascular diseases. One of the adequate strategies for the management of dysphagia is feeding people texture-modified food. This adaptation is determined by the nutritional needs, type, and level of dysphagia of those who suffer from it [1].

The development of new foods with textures adapted to dysphagia highlights the need to integrate knowledge from the clinical field with that of food technology [2]. Therefore, in recent years research has been carried out in which new foods are evaluated by people with dysphagia [3]. In these studies, people accept to a greater degree the modified foods with good flavor and nutritional contents, as well as foods that are easy to swallow. With such foods, patients would decrease malnutrition and dehydration, consequences of untreated dysphagia, and thus improve their quality of life [4]. In addition, the social and hedonic component of the act of eating should be considered. This act has evolved from merely nourishing oneself into a practice that improves the quality of people’s lives, both physically and psychologically [5].

There are several technological limitations to preparing dishes (culinary recipes with two or more ingredients) with a modified texture without compromising the composition and nutritional value of the non-adapted dishes. Among these limitations of the transformation processes are loss of flavor, inadequate texture, and loss of nutrients due to thermal treatment [3,6,7]. Flavor qualities are of great importance in the preparation of these dishes, since, if they maintain the characteristics of the non-adapted dish, they can improve the social component mentioned above. Likewise, studying the relationship between the sensory characteristics of texture-modified dishes and their hedonic evaluation could facilitate the development or improvement of new dishes [3]. Conventionally, a descriptive sensory profile completed by trained panels has been used to identify this relationship. Currently, faster and more economical methods are available, such as the check-all-that-apply (CATA) method. This method does not require trained judges and is sufficiently robust to obtain the profile of a wide range of products [8]. Furthermore, the CATA method is well suited to study the judges’ acceptance and the attributes that improve or worsen the acceptance scores [9]. 

Thus far, research on food texture modification has focused on studying the effects of hydrocolloids on food texture or on the hedonic response in simple liquids such as water or coffee [10,11]. Few studies have tested the acceptability of complex dishes adapted to dysphagia [3,12]. Additionally, the sensory characteristics of dishes adapted to dysphagia already accepted by the affected population have not yet been established. Therefore, it is necessary to evaluate the sensory attributes that can be objectively and quantitatively measured so that a texture-modified dish is organoleptically accepted by people with dysphagia. To this end, this work sets the following objectives: (i) to sensorially characterize texture-modified dishes for dysphagia associated with cerebral palsy (CP); (ii) to correlate these sensory characteristics with the degree of acceptance of the dishes scored by non-dysphagic judges; (iii) to compare the degree of acceptance of the texture-modified dishes between dysphagic and non-dysphagic judges; and (iv) to evaluate if the texture-modified dishes maintain the sensory characteristics of the non-adapted dishes.

## 2. Materials and Methods

### 2.1. Texture-Modified Dishes

The research was carried out with five dishes: chickpea stew (CS), lentils with rice (LR), chicken stew (ChS), halibut (*Hippoglossus hippoglossus*) with green sauce (H), and pasta bolognese (PB). According to the speech therapists of Spanish Confederation of Associations for the Care of People with Cerebral Palsy (ASPACE), in its provincial delegation of Navarre, ASPACE Navarra Foundation (Cizur Menor, Spain), these dishes are difficult to texturize. The recipes of the dishes were developed, and their cooking, the adaptation of their texture, and their cooling were carried out in the way described previously [12]. Thus, the present study was made to evaluate the sensory characteristics of dishes suitable for dysphagia. The dishes were made in the facilities ASPACE Navarra Foundation by the catering company Irigoyen Comedor Saludable S.L. (Pamplona, Spain). The dishes were stored at 4 ± 1 °C in a refrigerator (Liebherr, Ind Met S.A., Pamplona, Spain) until the moment of the sensory evaluation.

### 2.2. Sensory Evaluation

The sensory evaluations were carried out in the facilities of ASPACE Navarra Foundation and the Universidad Pública de Navarra (Pamplona, Spain) within the first 24 h after the preparation of the dishes. In the test conducted at the ASPACE Navarra Foundation, the dishes were evaluated in a room adapted to the good practices of sensory evaluation methodology, i.e., with appropriate light and space to avoid any interaction among the judges. At the university, the test was similarly carried out in a sensory evaluation room with individual booths [13].

Samples of 50 g of each of the texture-modified dishes were presented in disposable containers (Figure A1). The samples were weighed on a scale (EW 1500-2M, KERN & SOHN GmbH, Balingen, Germany) at the time of preparation. Samples were presented in monadic sequence, coded with three-digit random numbers, following a balanced Williams Latin square design order [14]. The samples were heated to 45 ± 5 °C in a microwave oven (WHITE, Lacor Menaje Profesional S.L., Guipuzkoa, Spain) for one minute before the sensory evaluation. This way, judges received heated samples before proceeding to the evaluation.

In this order, non-dysphagic judges were asked to give a hedonic evaluation, select descriptors following the CATA method, and identify each dish. Judges performed the sensory evaluation according to a protocol mimicking the oral processing of those with dysphagia associated with CP. This protocol consists of four phases: first, the spoon is filled to approximately half its capacity; second, the sample is placed in the middle of the tongue; third, the tongue is supported against the incisors, raised, and moved against the palate so that the sample does not disperse throughout the mouth; and fourth, the food bolus is pushed towards the back of the mouth to initiate swallowing.

Data collection for further processing was performed using Compusense^®^ Cloud software version 20.0.7464.31244 (Guelph, ON, Canada).

#### 2.2.1. Judges

Seventy non-dysphagic people participated in the evaluation (55 females, 15 males), ranging in age from 18 to 72 years. The gender ratio is representative of the Spanish population who work in health care. Women account for 84.0% of caregivers and social workers and 84.3% of nurses, according to the data of the Ministry of Education, Social Policy, and Sport of the Government of Spain [15,16]. These people were recruited from the ASPACE Navarra Foundation and the Universidad Pública de Navarra. All judges were non-dysphagic, because the CATA method is inappropriate for people with CP.

#### 2.2.2. Hedonic Evaluation

The first phase of the session consisted of a hedonic evaluation of the five dishes. This was done by means of a discrete, unipolar, nine-point numerical response scale, with 1 representing the lowest score and 9 the highest.

Hedonic acceptance of each dish was evaluated by non-dysphagic judges. The variables of sex and age (<24, 25–54, 55–64, and >65 years) were considered. The results of this hedonic evaluation were compared with those submitted by a panel of dysphagic judges with CP from a previous study [12]. To harmonize the scales of both panels, a categorization was made according to the following criteria: low level (equivalent to points 1, 2, and 3 of the numerical scale), medium level (equivalent to points 4, 5, and 6 of the numerical scale), and high level (equivalent to points 7, 8, and 9 of the numerical scale). Finally, for each dish, the judgments of the two panels were compared based on the proportion of judges from each panel who scored each of the three levels of acceptance.

#### 2.2.3. Selection of Attributes Using the CATA Method

The second phase of the session was a test using the CATA method [8] to select the sensory characteristics of adapted dishes and the relationship between attributes. Selection of attributes was based on a list of words generated by consensus of the test developers in two 1.5-h sessions. This list included 28 attributes, among which were terms describing different texture characteristics (“thick puree”, “lumpy”, “fibrous”, “floury”, “soft texture”, “liquid puree”), terms describing the behavior of the food during the oral phase and during swallowing (“sticky”, “leaves residue”, “bolus-forming”, “it spreads”, “easy-to-swallow”, “difficult-to-swallow”), terms relating to the original non-adapted dishes (“odor of the original dish”, “meat odor”, “fish odor”, “meat flavor”, “flavor of the original dish”, “fish flavor”), some descriptors that have been mentioned in the bibliography on texture-modified dishes (“metallic taste”), and some other descriptors of general characteristics of sensory quality (“strong odor”, “mild odor”, “strange odor”, “heterogeneous appearance”, “homogeneous appearance”, “strange flavor”, “bland”, “flavorful”, “salty”).

The attributes were grouped into four blocks according to the phases of swallowing: olfactory, visual, oral, and gustatory. Finally, the presentation of the attributes was balanced among the judges following a Williams Latin square design [14].

#### 2.2.4. Identification of the Original Non-Adapted Dishes

The third phase of the session consisted of identifying the original non-adapted dish (cooked only) from the texture-modified dishes given to the judges. To this end, judges were asked if they recognized the non-adapted dish (dichotomous “yes/no” question). In case of an affirmative answer, the name of the dish being identified was requested. The answers were classified into four levels: “correct”, correctly identified dish; “close”, similar dish or main ingredient perceived; “incorrect”, wrong answer; and “no response/do not know” (NR/DK). The cut-off point for correct identification was established at 62.5% by the formula (*n*+1) × 100/2*n*, with the number of levels *n* = 4.

To represent the descriptions the judges provided when attempting to identify the dishes, word clouds were generated from those attributes whose frequency of mention was higher than two, using the online software WordItOut (Enideo, Antwerpen, Belgium).

### 2.3. Statistical Analysis

The one-way analysis of variance followed by a Fisher’s least significant differences (LSD) test (*p* < 0.05) was performed on the hedonic evaluations to determine differences between texture-modified dishes. A Cochran’s Q test together with a Bonferroni test (post hoc multiple comparisons) [17] were applied to data obtained by the CATA method to analyze significant differences between dishes for each attribute. A multiple correspondence analysis (MCA) [18] was performed to explore the degree of relationship between dish samples and attributes. A principal component analysis (PCA) [19] was carried out to obtain the sensory profile of the texture-modified dishes. A penalty analysis was applied to study the effect of each attribute on the judges’ mean hedonic score. Finally, the comparison with a panel of dysphagic judges was performed by means of a comparison of *K* proportions. To this end, the independence between the two panels (dysphagic and non-dysphagic judges) and the three levels of acceptance was studied by means of Chi-square goodness of fit test (χ^2^) [17] and Monte-Carlo test [20], with a significance *p* < 0.05. A Monte-Carlo test was performed to simulate the real acceptance scores and to predict future scores. This simulation was carried out for both panels. With 5000 iterations, the test performs a statistical independence study, and it reinforces the Chi-square test (χ^2^) as it estimates the real scores. All statistical analyses were carried out using XLSTAT software version 2020.2.3. (Addinsoft, Paris, France).

## 3. Results

### 3.1. Hedonic Acceptance of Texture-Modified Dishes

The results of the hedonic evaluation are shown in Table 1. All dishes showed significant differences (*p* < 0.05) in this test. Neither age nor sex influenced the acceptability of these dishes. ChS and CS dishes had higher hedonic acceptance, with a score of M = 6.6 (SD = 1.7) and M = 6.4 (SD = 1.7), respectively. In contrast, LR was the dish that obtained the lowest score, M = 4.7 (SD = 2.3).

In the present work, hedonic scores of non-dysphagic judges were compared with those of dysphagic judges from previous studies [12]. These results are depicted in Table 2. The table shows that the results were very similar in both cases, the only differences being in the mean level of CS acceptance and in the three levels of PB acceptance. In the case of CS, the proportion of non-dysphagic judges who provided an intermediate score was higher than dysphagic judges (Appendix A
Table A1). In contrast, PB showed differences at all levels of acceptance. Proportionally, there were more dysphagic judges that gave high hedonic scores than the non-dysphagic judges, who gave more low and medium hedonic scores (Table A1).

### 3.2. Sensory Characteristics of Texture-Modified Dishes

The independence test between 28 attributes and five dishes was performed and showed that the attributes were significantly different (*p* < 0.05) for each dish. Therefore, the sensory characteristics were analyzed individually, and not for all of the dishes collectively. In other words, each dish possessed some unique characteristics.

A Cochran’s Q test was performed to assess the significant differences between dish samples for the 28 attributes included in the CATA method. The results are shown in Table 3. This table summarizes the sensory characteristics of each texture-modified dish. Among all dishes, 12 attributes were not significantly different (*p* > 0.05). Of these attributes, four stood out for their frequency of mention (more than 10 mentions for each dish) and served to define the general characteristics of all of them. These attributes were “homogeneous appearance” (visual phase), “easy-to-swallow” (oral phase), “odor of the original dish” (olfactory phase), and “flavor of the original dish” (gustatory phase). The 16 attributes that did show significant differences (*p* < 0.05) were used to characterize the different texture-modified dishes.

MCA facilitated the study of each attribute used to describe the sensory profile of the texture-modified dish samples. The 16 attributes used were grouped into three functions that explained 96.5% of the information. These functions were used to construct the symmetric graphs in Figure 1 and Figure 2 and to display the attributes selected to describe each dish. ChS stood out for its proximity to the attributes “flavorful”, “meat flavor”, and “meat odor”. CS was the dish closest to the attributes “soft texture” and “mild odor”. The H dish obtained the greatest number of responses with the attributes “fish odor” and “fish flavor”. PB was the dish closest to CS in the attribute “soft texture”, although it was also described by the judges as “sticky”, “strange flavor”, and “bland”. Finally, the attributes “strange odor”, “strange flavor”, “bland”, and “thick mash” were the most mentioned to describe the LR texture-modified dish.

A PCA was conducted to establish the relationships among the 16 attributes that helped interpret the sensory profile of texture-modified dishes. A correlation matrix was generated by PCA (Table A2). In this matrix, relationships between attributes are shown, direct (when an attribute increases, the other does too) and inverse (when an attribute increases, the other decreases). The direct relationships of “meat odor” and “meat flavor” with “metallic taste” were highlighted (degree of correlation ≥ 0.5). Similarly, the inverse relationships between the attributes “salty” and “sticky”, the attributes “meat odor” and “leaves residue”, and the attributes “metallic taste” and “leaves residue” were important (degree of correlation ≤ −0.5).

The most influential attributes in the hedonic evaluation and their relationships to the frequency of mention (expressed as a percentage) were considered. Attributes with a choice threshold higher than 20% and that also influence the hedonic acceptance score were chosen. Figure 3 and Figure 4 show the attributes that most influenced a positive score. The most outstanding attributes were “flavorful”, “flavor of the original dish”, “soft texture”, “easy-to-swallow”, and “odor of the original dish”. The attributes that negatively influenced the judges’ hedonic evaluation were “thick mash” and “bland”.

Figure 3 shows that some attributes with a higher percentage of selection were not significantly influential on the mean of the hedonic evaluation. This was the case for attributes such as “homogeneous appearance” or “mild odor”. Attributes with positive effects on the mean of the hedonic evaluation, such as “bolus-forming”, were also observed. Similarly, there were attributes with a negative effect such as “difficult-to-swallow”, “strange flavor”, or “liquid mash”.

### 3.3. Identification of Non-Adapted Dishes from Texture-Modified Dishes

The non-dysphagic judges were asked about the perceived dish when tasting the samples. Figure A2 shows the proportions of correct, close, incorrect, and NR/DK answers. The best-identified dish was H with 54 correct identifications. This success rate corresponded to 79% correct answers and 11% close approximations. This represents a success in the identification of this dish. It was followed by LR with 40 correct identifications (51%), PB with 27 (39%), ChS with 10 (14%), and CS with nine (13%). This is clearly a low level of recognition for all four latter dishes. Regarding the count of incorrect responses, CS was the dish with the highest number, with 33 (47%). The dish with the fewest incorrect answers was H with 0 (0%). PB was the dish with the most NR/DK responses with 26 (37%). The dish with the least number of NR/DK responses was H with seven (10%).

Figure A2 also shows the word clouds for each dish. The size of each word indicates the frequency of mention of the dishes or foods identified by the judges. In the case of H, the dish with the most correct answers, the word “fish” was mentioned up to 8–10 times more than other words such as “hake”, “cod”, or “ajoarriero” (Figure A2d). Similarly, for LR (Figure A2b), the most frequently mentioned word was “lentils”. Conversely, the dish with the least correct identifications was CS. It has as its most frequently mentioned word “chickpeas”. Other words such as “peppers” and “cod-casserole” were referenced with a similar frequency of mention as that of the main word (Figure A2a). Similarly, ChS (Figure A2c) was identified a greater number of times as “chicken”, although other words appear at practically the same frequency of mention, such as “peppers”. Finally, although the most mentioned word in PB (Figure A2e) was “pasta”, others such as “tomato pasta” or “tomato macaroni” were also frequent. 

## 4. Discussion

### 4.1. Hedonic Acceptance of Texture-Modified Dishes

Dishes adapted to dysphagia obtained medium-high hedonic scores (Table 1). The higher scores of CS and ChS could be due to their similarity in the “flavorful” attribute, in which both dishes stood out from the rest (Table 3). As shown by Hoppu et al. [21], the consumer is more willing to accept these products when the salt level is appropriate, because salt enhances the flavor of the dishes. In addition, penalty analysis (Figure 3 and Figure 4) confirmed that the “flavorful” attribute was a key factor in the acceptance of texture-modified dishes. ChS had the highest hedonic acceptance of all the five dishes. As for CS, it was distinguished by the attributes “soft texture” and “mild odor”. Both attributes stood out in the analysis of hedonic acceptance as two of the “nice to have” attributes (those that increase the hedonic acceptance of the dishes). It has been previously established that a low firmness [12] or a soft mouthfeel [22,23] are essential in dishes or foods intended for dysphagia. They should also be soft, smooth, creamy, and not grainy or sticky [24]. As for the attribute “mild odor”, the increase in the mentions of this attribute results in an increased hedonic acceptance of the dish (Figure 3 and Figure 4). This result is consistent with that obtained by Fondberg et al. [25], whose study concludes that the perception of flavor and hedonic acceptance increase when odor is not too strong but sufficient to evoke a pleasing memory.

In a second group, moderate hedonic acceptance of H (M = 5.8; SD = 2.1) could be due to the characteristic attributes of “fish odor” and “fish flavor”. As established by Cavicchi and Santini [26] for dishes and food in general, the perception of recognizable, traditional characteristics in food innovations could increase their hedonic acceptance. This effect could be extrapolated to dishes intended for dysphagia, since, in the current case, these dishes are a radical departure from the norm for non-dysphagic judges. Continuing, PB was the most prominent in the attribute “soft texture”, next to CS. A greater hedonic acceptance would be expected with these results. However, they also were assigned the attributes “sticky”, “strange flavor”, and “bland”. This latter attribute showed the greatest negative influence in the penalty analysis (Figure 3 and Figure 4), and thus could have decreased the hedonic acceptance of the dish. This is supported by Hoppu et al. [21], who explain that the lack of salt in dishes such as miso soup can lead to a decrease in the hedonic score of the dish. Similarly, other authors [12,27] had already established in dishes adapted to dysphagia that “sticky” is an undesirable attribute, and consequently that this feature should be avoided. The “strange flavor” detected in this dish could be due to food additives. It has been found that some food additives such as xanthan gum can alter the flavor profile of foods such as apple juice, even in amounts of 0.05 (*w*/*v*) [28]. In the present study, texturizers were used in amounts up to 1% (*w*/*w*), so it seems reasonable that the foreign flavor may come from them. Moreover, when usual or traditional flavors are not detected, hedonic acceptance could decrease 

Finally, LR obtained a score M = 4.7 (SD = 2.3), which could be equivalent to “neither like nor dislike”. This score could be because this dish with its modified texture was described by the attributes “strange flavor”, “bland”, and “thick mash”. The latter stood out in the penalty analysis as a “must not have” attribute, which decreased the hedonic acceptance of the dishes in which it appeared. This negative effect is confirmed by Hall and Wendin [22], who found that dishes that were “too thick” (firm), such as texture-modified meat or vegetable pies, are not accepted by elderly people with dysphagia or swallowing difficulties. Dysphagic individuals do not have sufficient motor capacity to deform or swallow firm or solid foods, so these foods must be modified [29].

Table 2 shows the results obtained with the Chi-square test (χ^2^) and Monte Carlo test to compare the hedonic evaluation of non-dysphagic judges (the present study) with that of CP-associated dysphagic judges (previous study, [12]). Except for CS (middle level) and PB (all levels), the results were consistent in both panels for all dishes and for all hedonic acceptance levels. CS yielded moderate acceptance by non-dysphagic judges (6.4/9) and high acceptance by dysphagic judges (1.5/3, with 1 as the best score). A similar response was also observed for PB. The panel of judges with dysphagia associated with CP showed a higher degree of hedonic acceptance (1.2/3, with 1 as the best score) than the panel of non-dysphagic judges (5.5/9). The other dishes were accepted at similar scores in both panels. LR showed the lowest acceptance, a finding attributable to its excessive firmness. The results obtained in the present work agree with those of Okkels et al. [3], in which dysphagic elders gave high hedonic scores (1.5/3) to the texture-modified products. Likewise, Lepore et al. [30] conclude that non-dysphagic adults, both young and old, prefer texture-modified dishes over those adapted and shaped to the original food. This is because shaped dishes are more difficult to swallow than texture-modified ones [24]. The dishes in the present work were only texture-modified (Figure A1), perhaps improving the hedonic acceptance in both panels.

The similarities between the two panels allowed for the study of the characteristics of the five dishes and the relationship between their attributes and hedonic acceptance. However, non-dysphagic people only studied the organoleptic characteristics and hedonic acceptance of suitable dishes, and they did not evaluate suitability characteristics. Therefore, the authors compared the results here presented with those of a previous study examining dish acceptance among dysphagic people [12], which led the authors to establish the suitability of the dishes analyzed in the present study. This comparison was fundamental to characterizing the texture-modified dishes.

### 4.2. Sensory Characteristics of Texture-Modified Dishes

Four attributes were frequently mentioned and showed no significant differences (*p* > 0.05) between dishes (Table 3). They were used to describe the general characteristics of the five dishes with texture modified. The attribute “homogeneous appearance” stands out. Homogeneity of texture has been referred to in the literature as an essential characteristic of products for dysphagia [31]. The present work concludes that not only texture but also a homogeneous appearance is of fundamental importance to dishes with texture-modified for dysphagia. “Easy-to-swallow” was the second attribute that describes all texture-modified dishes. The positive effect of this attribute on hedonic acceptance was demonstrated in the penalty analysis (Figure 3 and Figure 4). This is consistent with the literature, where it is referred to as “fluidity”, a prerequisite for dishes intended for dysphagia [12,22]. This ease of swallowing for texture-modified dishes was due to the effect of the food additives. These modify the mucoadhesivity (ability of food and dishes to adhere to the mucous membranes present during the swallowing process) of the dishes. Xanthan gum or guar gum changes the mucoadhesive strength of the dishes, improving the formation of the food bolus and its lubrication by the saliva. According to Cook et al. [32], this effect should be further studied in solid foods. However, in the current work these additives were effective in modifying the texture of complex dishes. Additionally, “flavor of the original dish” and “odor of the original dish” describe the characteristics of dishes for dysphagia. Both were important in the penalty analysis and designated “nice to have” (Figure 3 and Figure 4). The importance of perceiving traditional characteristics in food innovations has already been stated by Cavicchi and Santini [26]. It is important that these attributes or characteristics are easily recognized, as in the present study, since people with dysphagia associated with CP have altered thresholds of taste sensitivity [33]. Not being able to recognize the non-adapted dish via taste or odor when presented with the adapted dish would reduce its hedonic acceptance [34].

After the study of the general characteristics of the texture-modified dishes, the characteristics of each dish were evaluated separately. For the MCA, the 16 attributes associated with the texture-modified dishes (*p* < 0.05) according to Cochran’s Q test were used. The MCA revealed the characteristics of each dish, as well as the similarities and differences between them, as shown in Figure 3 and Figure 4. The two meat dishes were more often described with attributes referring to the characteristics of their non-adapted dish. Thus, ChS was associated with the attributes “meat odor” and “meat flavor”, and H was the most linked to “fish odor” and “fish flavor”. In the case of H, the processing of fish, and especially its baking―as in the recipe used in this study―caused an increase of odor and taste characteristics [35]. The attribute “mild odor” served to differentiate CS from the rest of the dishes. In addition, along with ChS, the attribute “flavorful” stood out, with both attributes especially valued (Figure 3 and Figure 4). That the dishes present not-too-strong and recognizable odors helps to increase the perception of flavor [25]. The right amount of salt increases the acceptability of the dish [36] and even swallowing safety for people with dysphagia [32,37]. For its part, PB was often assigned the “soft texture” attribute, similar to CS. This could differentiate it as a dish more suitable for dysphagia, since low firmness is essential for people with this condition to have greater ease in swallowing [12,22,23]. However, PB also stood out in attributes such as “sticky,” which should not appear in products intended for dysphagia [12], and “bland,” which has been shown to reduce the acceptance of dishes [21,36]. This latter attribute was shared by LR. This dish stood out from the rest in having the attributes of “strange odor” and “strange flavor”, characteristics that can come from the texturizing agents. Some of them, such as xanthan gum, confer strange flavors or diminish the flavor of the dish to which they are added [28]. As shown in Figure 2, this dish was associated with the attribute “thick mash”, which has negative effects on hedonic acceptance according to dysphagic judges. This is consistent with what was previously called “excessive firmness” [12].

The results of the PCA analysis for the associations between attributes are shown in Table A2. The direct relationship between the attributes “meat odor” and “meat flavor” with “metallic taste” was evaluated. This relationship was confirmed in ground beef or lamb mixed with other ingredients, but was weak in chicken [38]. It is attributed to the presence of minerals such as calcium, magnesium or iron, which can provide taste after cooking [39]. The attributes “salty” and “flavorful” influence one of the most important attributes to consider for the preparation of dishes for dysphagia, “sticky”. As Choichuedee et al. [40] established through rice-based dishes and sensory evaluation by dysphagic elderly, “sticky” decreases the hedonic acceptance of dishes. Salt is a fundamental ingredient since it stimulates a greater secretion of saliva during the chewing of food. The greater the amount of salt in the food, the greater the amount of saliva produced during chewing, favoring the generation of the food bolus and the lubrication of the food during swallowing [32,37]. It is for this reason that the attributes “flavorful” and “salty” are inversely related to the attribute “sticky”. Finally, the inverse relationships between “meat odor”, “meat flavor”, and “metallic taste” with “leaves residue” were evaluated. These evaluations were studied simultaneously, because the relationship between the “meat odor/flavor” and the “metallic taste” had already been established. The inverse relationship between these attributes can be given, because chicken is meat, and although it is not fatty, it can increase the juiciness and smoothness of the dish when processed [41]. This can be augmented by the recipe of the dish, which in addition to high chicken content (56.5%) contains olive oil (1.4%) and its lubricating characteristics. This greater lubrication and juiciness can diminish the stickiness of the dish and, therefore, the inverse relationship between these attributes.

### 4.3. Identification of Non-Adapted Dishes from Texture-Modified Dishes

As mentioned above, people with dysphagia associated with CP have altered thresholds of taste sensitivity [33]. For this reason, and to bring these people closer to the important social act of eating [5], it is necessary for texture-modified dishes to be easily recognizable. Additionally, they must have characteristics of their non-adapted dishes. To know if the texture-modified dishes of the current work had these characteristics, their identification was asked of the judges. Figure A2 shows the judges’ success rate in predicting the dish they tasted (left) and the word clouds of the five dishes (right). These word clouds provide visually useful information to see the judges’ perceptions [42]. The only dish that had a clear success in its identification was H, with 54 correct identifications out of the possible 70 (Figure A2g). The increased flavor of the processed fish [35] made it easy to identify. In addition, the alternative characteristics in its word cloud are scarce and correspond to white fishes such as “hake” or “cod” (Figure A2h). The rest of the dishes had a low identification level, with LR having the highest percentage of correct identifications with 36 (51%) (Figure A2c). Lentils, like fish, increase their flavor when processed [43], which could contribute to part of this successful identification. The frequency of mention of the characteristic “lentil” stands out clearly over characteristics such as “legume vegetables” or “kidney beans”, which were part of the alternatives identified. PB had only 27 correct identifications (Figure A2i). This may be due to the amount of tomato in its recipe (28%), which makes the frequency of mention of dishes such as “tomato pasta” or “tomato macaroni” more notable (Figure A2j). For the ChS dish, the number of correct identifications dropped to 10 (Figure A2e). It has become evident that chicken brings little flavor and even diminishes the existing flavor in dishes where it is added [41]. This explains why a large proportion of the ingredients in its recipe was found by judges (Figure A2f), as in the case of “stuffed peppers”. The dish with the least correct identifications was CS with 9 (Figure A2a). Dandachy et al. [44] stated that chickpeas lose flavor when processed and made into flour, and the same could happen with chickpeas after cooking and grinding. Some of the ingredients of this dish [12] can be recognized as “peppers” or “vegetables” with a large frequency of mention (Figure A2b). These flavors could be identified because peppers and vegetables were part of the chickpea recipe. This decrease in the flavor and the fact that the dish was cooked with vegetables could complicate the identification of this flavor, which made its number of correct identifications the lowest.

Except for H, the non-adapted dishes were difficult to identify from the texture-modified dishes. It has been shown that the acceptance of simple adapted dishes such as purees significantly decreases when compared to non-adapted dishes from which they originate [34]. This may be due to the loss of sensory characteristics of dishes, as noticed in this work. It is important to increase their characteristics to achieve the greatest possible similarity of these texture-modified dishes to their non-adapted counterparts.

In the current study, the feasibility of the CATA method to characterize the texture-modified dishes for dysphagia associated with CP has been proven. However, this research has some limitations. The findings can only be applied to the five selected dishes of this study. Additionally, results cannot be extrapolated to dysphagia associated with other pathologies. Different dishes should be evaluated by people with dysphagia associated with other causes.

## 5. Conclusions

The CATA method made it possible to establish the sensory characteristics of dishes adapted to dysphagia associated with CP. For greater acceptance, these dishes must show a homogeneous appearance, be easy to swallow and flavorful, and have a soft texture. These attributes improve the evaluation of non-dysphagic people. Dishes with little flavor and which are too firm or thick receive lower hedonic scores. Non-dysphagic judges have found these attributes to be important in evaluating texture-modified dishes either positively or negatively. The only non-adapted dish correctly identified from its texture-modified dish was fish in green sauce. The remaining dishes had low success rates of identification. 

More studies are needed to evaluate other attributes that have not been identified in this work and that could contribute to the suitability and organoleptic characteristics (desirable or undesirable) of dishes intended for dysphagic people. In addition, it will be necessary to investigate what changes in processing or in recipes will have to be made to achieve greater similarity to the original non-adapted dishes upon which dishes adapted to dysphagia associated with CP are based.

## Figures and Tables

**Figure 1 foods-10-00480-f001:**
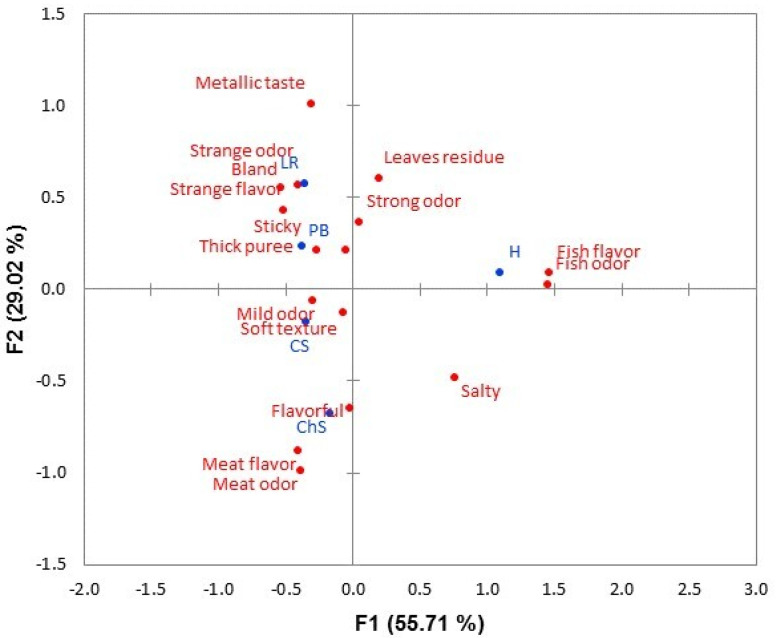
Symmetric graph of attributes and dishes (obtained by multiple correspondence analysis) for the sensory differences between texture-modified dishes to CP dysphagia. Dishes: chickpea stew (CS), lentils with rice (LR), chicken stew (ChS), halibut with green sauce (H), and pasta bolognese (PB).

**Figure 2 foods-10-00480-f002:**
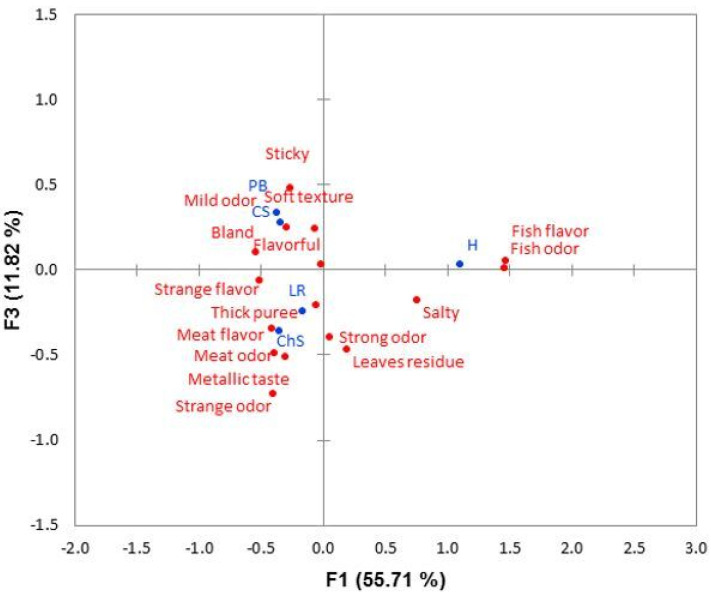
Symmetric graph of attributes and dishes (obtained by multiple correspondence analysis) for the sensory differences between texture-modified dishes to CP dysphagia. Dishes: chickpea stew (CS), lentils with rice (LR), chicken stew (ChS), halibut with green sauce (H), and pasta bolognese (PB).

**Figure 3 foods-10-00480-f003:**
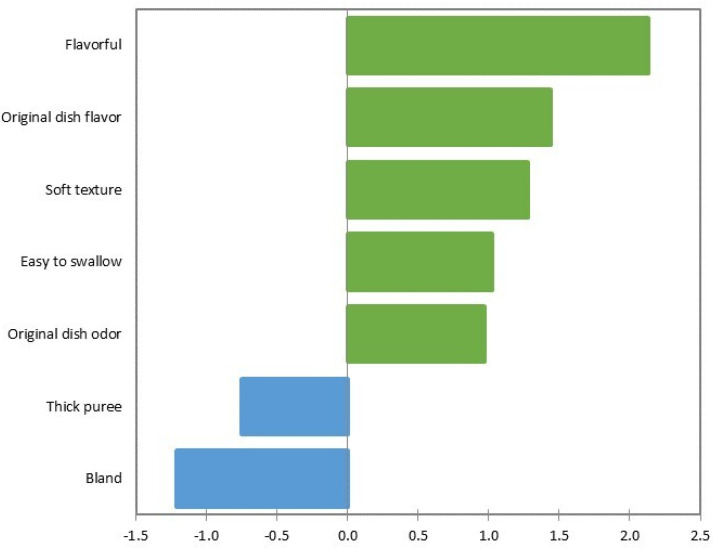
Effect of the CATA method attributes on judges’ hedonic evaluation about the texture-modified dishes for CP dysphagia. Mean drops vs. percentage chart to identify “must not have” and “nice to have” attributes with significant effect (*p* < 0.05) on the mean.

**Figure 4 foods-10-00480-f004:**
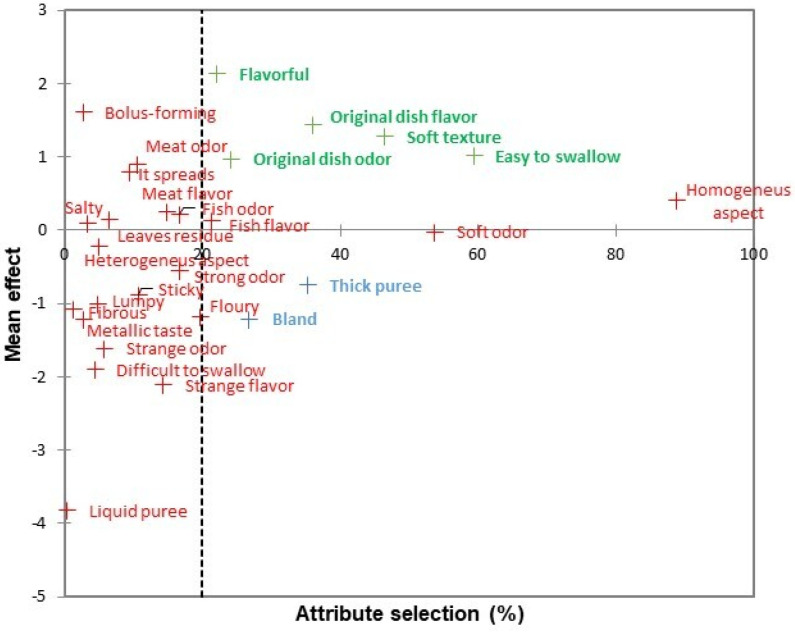
Effects on the mean and the percentage of attributes in the judges’ hedonic evaluations about the texture-modified dishes. Blue words signify “must not have” attributes, and green words identify those that are “nice to have” with significant effect (*p* < 0.05) on the mean.

**Table 1 foods-10-00480-t001:** Hedonic scores (Mean and SD) of the texture-modified dishes evaluated by the non-dysphagic judges (*n* = 59).

Dish	Hedonic Score
Chickpea stew	6.4 (1.7) ^a,b^
Lentils with rice	4.7 (2.3) ^d^
Chicken stew	6.6 (1.7) ^a^
Halibut with green sauce	5.8 (2.1) ^b,c^
Pasta bolognese	5.5 (2.1) ^c^

Different superscripts for each dish differ significantly (*p* < 0.05) by the Fisher’s LSD test.

**Table 2 foods-10-00480-t002:** Results of the Chi-square (χ^2^) and Monte-Carlo tests, according to the acceptance levels, to detect possible differences between the non-dysphagic and dysphagic panels.

Hedonic Acceptance Level	Test	Chickpea Stew	Lentils with Rice	Chicken Stew	Halibut with Green Sauce	Pasta Bolognese
Low	χ^2^	0.081	0.194	0.151	0.786	0.054
	Monte-Carlo	0.110	0.212	0.259	1.000	0.095
Medium	χ^2^	0.014	0.064	0.263	0.136	0.004
	Monte-Carlo	0.017	0.086	0.285	0.163	0.008
High	χ^2^	0.223	0.556	0.108	0.104	<0.0001
	Monte-Carlo	0.294	0.661	0.143	0.124	<0.0001

The degree of acceptance differs between panels if the difference between values is significant (*p* < 0.05).

**Table 3 foods-10-00480-t003:** Citation frequency of attributes associated with CP dysphagia adapted dishes in the CATA method.

Phase	Attributes	Chickpea Stew	Lentils with Rice	Chicken Stew	Halibut with Green Sauce	Pasta Bolognese	*p*-Value
Olfactory	Strong odor	3 ^b^	19 ^a^	9 ^a,b^	14 ^a,b^	7 ^a,b^	0.000
	Mild odor	55 ^a^	29 ^b,c^	33 ^b^	18 ^c^	44 ^a,b^	<0.0001
	Original dish odor	13	18	17	21	21	0.263
	Strange odor	2 ^b^	12 ^a^	3 ^b^	1 ^b^	4 ^b^	0.002
	Meat odor	6 ^b^	3 ^b^	25 ^a^	1 ^b^	2 ^b^	<0.0001
	Fish odor	1 ^b^	2 ^b^	8 ^b^	47 ^a^	2 ^b^	<0.0001
Visual	Heterogeneous aspect	1	6	4	6	1	0.205
	Homogeneous aspect	66	60	61	58	64	0.130
Oral	Thick puree	18 ^b^	39 ^b^	19 ^b^	21 ^a,b^	19 ^b^	0.001
	Lumpy	1	6	3	6	2	0.259
	Fibrous	1	3	0	1	0	0.075
	Floury	12	23	15	1	8	0.059
	Soft texture	40 ^a^	17 ^b^	34 ^a,b^	32 ^a,b^	39 ^a^	0.000
	Liquid puree	0	1	0	0	0	0.406
	Sticky	4 ^b^	5 ^a,b^	4 ^b^	4 ^b^	19 ^a^	0.000
	Leave residue	2 ^b^	9 ^a^	1 ^b^	6 ^a^	1 ^b^	0.006
	Bolus-forming	3	2	3	0	1	0.216
	It spreads	5	3	5	8	8	0.406
	Easy to swallow	44	32	42	45	32	0.086
	Hard to swallow	0	6	4	1	5	0.113
Gustatory	Metallic taste	0 ^b^	5 ^a^	0 ^b^	1 ^b^	2 ^b^	0.030
	Meat flavor	9 ^b^	4 ^b^	30 ^a^	0 ^b^	5 ^b^	<0.0001
	Original dish flavor	24	29	23	28	24	0.719
	Fish flavor	2 ^b^	3 ^b^	8 ^b^	60 ^a^	2 ^b^	<0.0001
	Strange flavor	8 ^a,b^	17 ^a^	5 ^a,b^	1 ^b^	15 ^a^	0.000
	Bland	19 ^b^	35 ^a^	3 ^c^	2 ^c^	31 ^a,b^	<0.0001
	Flavorful	24 ^a,b^	2 ^d^	31 ^a^	13 ^b,c^	6 ^c,d^	<0.0001
	Salty	0 ^b^	0 ^b^	4 ^a,b^	7 ^a^	1 ^b^	0.021

*p*-Value < 0.05 indicates differences in the attributes for each dish by Cochran’s Q test. Different superscripts in the same row correspond to the differences detected by Bonferroni test.

## Data Availability

Not applicable.

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
