# Peer review of "Dishes Adapted to Dysphagia: Sensory Characteristics and Their Relationship to Hedonic Acceptance"

_foods, 2021, doi:10.3390/foods10020480_

Round 1
Reviewer 1 Report
I found this paper rather wordy.
It is more common to use the term "texture modified" rather than "texture adapted". Is there a reason the authors have chosen the latter?
It is interesting that you use Halibut as a fish - is there a reason? I notice in the word clouds halibut does not come up, but terms like cod and hake occur. Isn't halibut rather expensive for this kind of a study?
Some other odd terms seem to occur such as "soft odor" how is was this defined, how were the assessors trained for it? Also use of the word "strange" - this seems a subjective term - were assessors given a "strange" standard food or might it be a different stimulus for each assessor? Moreover figure 4 includes attributes like "it scatters", yet this does not seem to be mentioned elsewhere, can you say anything about it?
Is "bland" a real attribute, or just something to describe the absence of everything else.
Dysphagia is quite a broad term and deals with individuals who have a range of swalloing difficulty. It is not clear, though I think from this paper that you are dealing with patients who have cerebral palsy. Is this the case? Please can you make it more obvious - perhaps adding it to the title. I am concerned about ethics of dysphagic patients. Certainly the study makes reference to non dysphagic assessors and their ethics - but it also refers to the authors previous publication Innovative Food Science & Emerging Technologies 2020, 64, 102383 for the dysphagic work - and this paper makes no mention of ethics in relation to recruiting the dysphagic assessors!!
I know that figure A1 is not strictly part of the paper, but were samples provided in this diverse range of dishes? While I can see you have random codes on the dishes the dishes themselves might influence the assessors.
Author Response
The attached file contains the answers.

Reviewer 2 Report
General comments
Texture modification of foods for dysphagia is a topic that is important due to the implication on both the health and nutritional status of individuals. There are a number of studies for example, a recent review on dysphagia by Munialo, et al (2020) on Rheological, tribological and sensory attributes of texture‐modified foods for dysphagia patients and the elderly which show sensory to be a key determinant on the acceptance of texture modified foods. Thus, the current work on the sensory characterisation of different texture modified foods for dysphagia is important; as this would help in broaden the number of foods that would be included within the IDDSI Framework. Thus, there is merit in the current work. However, there are a number of issues in the manuscript that would need to be addressed before the paper can be accepted for publication, which are listed below in addition to a thorough proofreading of the manuscript being carried out, and the grammatical errors revised. I would also like to see the authors discuss their current findings in the light of the IDDSI. The texture modified foods that they studied, in which category of the IDDSI framework would they fall?
Introduction
L40: please rewrite the sentence “With such foods, patients would avoid the consequences of untreated dysphagia, malnutrition, and dehydration, and thus improve their quality of life” to show that malnutrition, and dehydration as some of the consequences of untreated dysphagia.
L55, please revise
Materials and methods
L83, what does they mean in this context?
L84, Why were the dishes stored at 5 °C and not the typical 4 °C that is often used in chilling of products?
L90, what does a room readied mean?
L93 – 94, am not just sure why the authors mention that the room was at room temperature? I would imagine that if the judges tasted the foods within the sensory booths in a sensory lab, it would be obvious that the room was at room temperature.
L95 clearly the containers that were used in this study are different as illustrated in Fig.A1. Could the authors provide some information as to whether they weighed in to the containers the 50 g of sample before presenting the same to the judges?
L97 were the sample re-heated first before being given to the judges or were they presented to the judges then re-heated, as this does seem to be the case as outlined in the current description.
L172-174, please revise and use the correct tense
I am not sure about the experimental design for this study. L112 the authors state that the judges were non-dysphagic, then in lines 121, they say that the results were compared to previous studies that involved dysphagic judges. How can the authors be confident that the conditions and the food samples in the present study are the same as the published data? Why was there such a big variation in gender for the participants that were recruited (55 females and 15 males?)
Results and Discussion
L191-195, could the authors please show the data that confirms the fact that the proportion of non-dysphagic judges who provided an intermediate score was higher than dysphagic judges? This is key information for this study and needs to be added to the manuscript.
L204 Could the authors please change characteristics all its own to characteristics on their own?
L338 Please change find to found
L361-362 Please revise the following sentence “they were not studied the suitability characteristics”
L363 Please provide a reference to the previous study that the authors refer to.
Conclusion
More studies are only mentioned in the conclusion and not anywhere else in discussion. There should not be new aspects being introduced in the conclusion as this should be providing a summary of what was discussed in the paper.
Author Response
The attached file contains the answers.

Reviewer 3 Report
Very interesting work developed by Ibañez et al. The study is well designed and structured and its presentation has a very high level.
I only encourage the authors to mention the study limitations at the end of Discussion section.
Author Response
The attached file contains the answers

Round 2
Reviewer 1 Report
In what language did the assessors undertake the experimental work? If Spanish, might the terms you use in the paper be the result of your translation into English and would this give rise to your ability to change “soft” to “mild” in this revision? And now I notice the word smooth on line 18 of the abstract has become soft. Thank you for explaining “it scatters” now I see it has been substituted for “it spreads” and “it disperses” – though as a texture term “spreads” sounds more meaningful in English than “scatters”. If choice of these words is merely your choice as authors in attempting to translate the terms which the assessors used (in Spanish) then I suggest you employ a fluent English speaker with a good grasp of texture vocabulary (I would suggest you approach B Piqueras-Fiszman or M Aguayo-Mendoza) – I apologise for questioning your own fluency, but the terms you are using do not seem to read well for me (I am a native English speaker with no knowledge of the Spanish language, though I consider myself well versed in both instrumental and sensory texture testing).
I do not accept your distinction of texture adapted as opposed to modified. From my knowledge of the subject, texture adapted is not commonly used in the literature, whereas texture modified is widely used and in many cases refers to particular needs – see for example Chichero (2015) “Texture-modified foods for hospital patients” in Chen & Rosenthal (Eds) Modifying Food Texture v2 Sensory Analysis, Consumer Requirements and Preferences. Woodhead Publishing. I cannot see any of your references using the word adapted in their title – yet modified comes up in several. From what you say adapted is a subset of modified, if you are insistent on using “adapted” – can you give a reference in the text to identify where this definition arises?
I now see that I misunderstood your study and that you only undertook the data collection with non-dysphagic assessors. So I accept that my questions about ethics were inappropriate.
Author Response
Dear Reviewer,
Thank you very much for your helpful comments. The authors have revised the paper accordingly and want to highlight that your comments contribute to improve the manuscript. Please find our response (in blue) to specific comments below.
Comments and Suggestions for Authors
In what language did the assessors undertake the experimental work? If Spanish, might the terms you use in the paper be the result of your translation into English and would this give rise to your ability to change “soft” to “mild” in this revision?
The assessor used Spanish to carry out their tasks. A translation and edition company with experience in scientific English later translated the generated vocabulary.
And now I notice the word smooth on line 18 of the abstract has become soft. Thank you for explaining “it scatters” now I see it has been substituted for “it spreads” and “it disperses” – though as a texture term “spreads” sounds more meaningful in English than “scatters”. If choice of these words is merely your choice as authors in attempting to translate the terms which the assessors used (in Spanish) then I suggest you employ a fluent English speaker with a good grasp of texture vocabulary (I would suggest you approach B Piqueras-Fiszman or M Aguayo-Mendoza) – I apologise for questioning your own fluency, but the terms you are using do not seem to read well for me (I am a native English speaker with no knowledge of the Spanish language, though I consider myself well versed in both instrumental and sensory texture testing).
To enhance the comprehension of manuscript, the authors have changed “scatters” to “spreads”. The authors agree with your contributions to vocabulary of the sensory evaluation. Therefore, it is not necessary to consult an English speaker. The authors take into account recommended experts to collaborate in future studies.
I do not accept your distinction of texture adapted as opposed to modified. From my knowledge of the subject, texture adapted is not commonly used in the literature, whereas texture modified is widely used and in many cases refers to particular needs – see for example Chichero (2015) “Texture-modified foods for hospital patients” in Chen & Rosenthal (Eds) Modifying Food Texture v2 Sensory Analysis, Consumer Requirements and Preferences. Woodhead Publishing. I cannot see any of your references using the word adapted in their title – yet modified comes up in several. From what you say adapted is a subset of modified, if you are insistent on using “adapted” – can you give a reference in the text to identify where this definition arises?
Regarding “texture-adapted”, the authors have based on paper of der Stelt et al (2020). However, authors accept the suggestion and have replaced "texture-adapted" by "texture-modified".
Van der Stelt, A.J.; Mehring, P.; Corbier, C.; van Eijnatten, E.J.M.; Withers, C. A “mouthfeel wheel” terminology for communicating the mouthfeel attributes of medical nutrition products (MNP). Food Quality and Preference 2020, 80, 103822, doi:10.1016/j.foodqual.2019.103822.
I now see that I misunderstood your study and that you only undertook the data collection with non-dysphagic assessors. So I accept that my questions about ethics were inappropriate.
The authors appreciate the comment.
